# Fabrication and Characterization of Complex Coacervation: The Integration of Sesame Protein Isolate-Polysaccharides

**DOI:** 10.3390/foods12193696

**Published:** 2023-10-08

**Authors:** Zeng Dong, Shirong Yu, Kefeng Zhai, Nina Bao, Marwan M. A. Rashed, Xiao Wu

**Affiliations:** 1School of Biotechnology and Food Engineering, Suzhou University, Suzhou 234000, China; dongzeng@ahszu.edu.cn (Z.D.); 18715454293@163.com (S.Y.); ninabao@ahszu.edu.cn (N.B.); marwanrashed6@ahszu.edu.cn (M.M.A.R.); szxywx@ahszu.edu.cn (X.W.); 2Engineering Research Center for Development and High-Value Utilization of Genuine Medicinal Materials in North Anhui Province, Suzhou University, Suzhou 234000, China; 3College of Food Science and Technology, Nanjing Agricultural University, Nanjing 210095, China

**Keywords:** sesame protein isolate, biopolymers, complex coacervation microstructure, emulsion techno-parameters

## Abstract

The exceptional biocompatibility of emulsion systems that rely on stabilizing protein–polysaccharide particles presents extensive possibilities for the transportation of bioactive carriers, making them highly promising for various biological applications. The current work aimed to explore the phenomenon of complex coacervation between sesame protein isolate (SPI) and four distinct polysaccharides, namely, Arabic gum (GA), carrageenan (CAR), sodium carboxymethyl cellulose (CMC), and sodium alginate (SA). The study objective was achieved by fabricating emulsions through the blending of these polymers with oil at their maximum turbidity level (φ = 0.6), followed by the measurement of their rheological properties. The turbidity, ζ-potential, and particle size were among the techno-parameters analyzed to assess the emulsion stability. The microstructural characterization of the emulsions was conducted using both transmission electron microscopy (TEM) and scanning electron microscopy (SEM). Furthermore, the functional properties were examined using Fourier-transform infrared spectroscopy (FTIR) and X-ray diffraction (XRD). The SPI incorporated with SA, CMC, and CAR reached the maximum turbidity (0.2% *w*/*v*) at a ratio of 4:1, corresponding to the pH values of 4.5, 3, or 3.5, respectively. The SPI–GA mixture exhibited the maximum turbidity at a ratio of 10:1 and pH 4.5. Results from the FTIR and XRD analyses provided evidence of complex formation between SPI and the four polysaccharides, with the electrostatic and hydrogen bond interactions facilitating the binding of SPI to these polysaccharides. SPI was bound to the four polysaccharides through electrostatic and hydrogen bond interactions. The SPI–CMC and SPI–SA emulsions were more stable after two weeks of storage.

## 1. Introduction

There has been great interest recently in protein-rich crops and protein-based products. Protein-rich crops have potentially positive effects in reducing the risk of metabolic syndrome, regulating body weight and diabetes, and preventing cancer [1,2]. *Sesame indicum* L. is a significant oil crop abundant in oils and proteins [3]. The protein content (%) in sesame crops varies depending on factors such as the geographical region, environmental conditions, and growing season. In a study by Morris et al. [4], it was observed that the protein content (%) in 10 sesame cultivars grown in Sudan ranged between 33 and 40%, whereas in Turkey, it ranged from 3 to 25% with 35 sesame cultivars. Sesame cake possesses diverse biological properties, including anti-oxidation, anti-cancer, anti-inflammatory, and anti-mutation activities, as well as the ability to reduce blood lipid levels and blood pressure [5]. The majority of sesame cakes are utilized for feed and fertilizer purposes; a portion of them are disposed of as industrial waste.

Most polysaccharides in nature are negatively charged over a wide pH range, whereas proteins are positively charged below their isoelectric points (PI). Complex coacervation is often divided into two types based on the properties of the complex. When a complex is a liquid (droplet), it is called coacervation, and the fractal aggregate (solid) is called a complex [6]. Protein–polysaccharide interactions have an important impact on the functional properties of their mixtures. The interaction strength is usually affected by many factors, such as the pH, biopolymer ratio, concentration, polymer molecular weight, temperature, and medium ionic strength [6,7]. Complex coagulation is the embedding and controlled release of bioactive materials commonly used by researchers in agriculture, food, cosmetics, medicine, and chemical processing [8]. Moreover, proteins and polysaccharides combine into several products, contributing to their structure, texture, and stability [9]. The mechanism of the interaction between proteins and polysaccharides has been widely studied.

The complexes formed by proteins and polysaccharides perform better in stabilizing high internal phase emulsions because of the polysaccharide’s solid self-association property and the protein’s amphiphilicity [10]. Combining CMC and wheat protein can yield a stable O/W high internal phase emulsion. The rheological properties of emulsions can be adjusted by varying the degree of substitution of CMC. An increased internal phase emulsion with an oil volume fraction of 90.2% was prepared by homogenization centrifugation at 0.3%–1.5% (*w*/*v*) of gliadin–CMC composite particles. The emulsion exhibited good centrifugation, thermal, and storage stability. The negatively charged okara protein and CMC complex formed by the electrostatic interactions could reduce the oil droplet size and improve the stability of oil–water emulsions under acidic conditions [11]. They can affect the foaming properties of proteins to varying degrees. The emulsion has a sizeable internal phase ratio, excellent stability, and a uniform void of template materials with several applications [12].

Proteins are generally extracted using an alkali solution and acid precipitation. The sesame protein isolate can be used as a natural surfactant to enrich the ω-(3) preparation of PUFA nanoemulsions [13]. Saatchi et al. [14] found that the emulsification properties of the sesame protein–maltodextrin conjugates were better than those of the sesame protein alone. There are few reports on preparing the complex coacervation of sesame proteins and polysaccharides.

The current study will investigate the stability of four polysaccharides commonly found in the market, namely, Arabic gum, carrageenan, sodium carboxymethyl cellulose, and sodium alginate, when incorporated into a sesame protein isolate system. Various characteristics such as the potential, particle size, maximum turbidity, FTIR, XRD, TEM, SEM, and rheological properties of the emulsions formed by these complexes were assessed at different pH values. The primary aim of this study is to present significant findings and perspectives that can aid in the choice of polysaccharide stabilizers and the application of sesame protein in both the food and non-food sectors.

## 2. Materials and Methods

### 2.1. Materials

The sesame meal was purchased from a local market in Suzhou (China). CMC (600–1000 mpa·s, USP grade) was purchased from MACKLIN Chemistry Co., Ltd. (Shanghai, China). SA (MW 198.11 kDa, CP grade) was purchased from Sinopharm Chemical Reagent Co., Ltd. (Shanghai, China). CAR was purchased from Shanghai Biotechnology Co., Ltd. (Zhengzhou, China). GA (MW 262.64, Pharmaceutical Grade) was purchased from Aladdin Chemistry Co., Ltd. (Shanghai, China). Analytical-grade petroleum ether, sodium hydroxide, and hydrochloric acid were obtained from Sinopharm Chemical Reagent Co., Ltd. (Shanghai, China).

### 2.2. Preparation of Sesame Protein Isolate (SPI)

The sesame meal was crushed and degreased for 6 h using Soxhlet extraction. The protein content of the defatted sesame meal powder was 54.84% (N × 5.30), as determined by the Kjeldahl method. The defatted sesame meal powder was mixed with water at a ratio of 1:14, and the pH was adjusted to 12 using 0.5 M NaOH. The solution was heated in a 55 °C water bath for one hour, suction filtered, and then centrifuged (Thermo Lynx 4000, Thermo Scientific, Madison, WI, USA) at 5000 rpm for 20 min to obtain a clear protein solution. The pH of the solution was adjusted to 4.5, and the precipitate was obtained after centrifugation at 8000 rpm for 10 min. Finally, the precipitate was washed with water, neutralized, and freeze-dried using a lyophilizer (CVC FD8-6, SIM International Group, California, CA, USA) to obtain a pure sesame protein isolate (protein 90.53%, N × 5.30), ground using a mortar, and stored in a brown reagent bottle at 4 °C.

### 2.3. Solubility of SPI

According to [15], protein solubility was carried out with some modifications. Briefly, the protein solution was prepared at a concentration of 1% (*w*/*v*), and the pH was adjusted to 8 for complete dissolution. The protein solution was divided into ten parts. The pH was changed from 11 to 2 using a 1 M hydrochloric acid and 1 M sodium hydroxide solution. After adjusting the pH, the solution was centrifuged at 4500 r/min for 20 min to separate the insoluble fraction. The protein content of the solution was determined using the Coomassie Brilliant Blue method [16]. A standard curve was constructed using bovine serum protein (R^2^ = 0.9946), and the protein content in the solution was calculated as y = 0.0064x + 0.018, y is the absorbance value, and x is the protein content (µg/mL).

### 2.4. Turbidimetric Titration

Polysaccharides (GA, CAR, SA, and CMC) were dissolved in deionized water to prepare 0.2% (*w*/*v*) polysaccharide stock solutions. SPI was also dissolved in deionized water and stirred intensely with a high-shear homogenizer, and the pH was adjusted to 8 to dissolve all the proteins. The protein and polysaccharide solutions were mixed in different ratios to prepare a solution with a total solids content of 0.2% (g/mL). The solution was placed in a refrigerator at 4 °C overnight to hydrate thoroughly, and the pH was adjusted to prepare a protein–polysaccharide complex system with different pH values. The final product was stored at 4 °C until further use. When the pH of the mixed solution is >4.5, the pH is adjusted with δ-glucolactone; when the pH is >3.5, the pH with 0.05 M hydrochloric acid; when the pH is >2.5, the pH with 0.5 M hydrochloric acid; when the pH is ≥2.0, the pH is adjusted with 1 M hydrochloric acid [17]. The pH of the mixed solution was adjusted from the original value to 2.0, and the absorbance of the supernatant at 600 nm was measured with a pH reduction of 0.5. Deionized water was used as a blank, and the protein mother solution was used as a control.

### 2.5. ζ-Potential Analysis

Dynamic light scattering (Malvern Zeta Nano ZS90, Malvern Instruments, Worcestershire, UK) was used to measure the particle size and ζ-potential of the protein–polysaccharide mixture [18]. The ζ-potentials of SPI and polysaccharides at different pH values are also shown. The protein–polysaccharide mixture was diluted 100 times in pH-buffered deionized water, and the particle size was measured. All the measurements were performed using freshly prepared samples at angles of 20° and 90°.

### 2.6. Coacervate Yield of Concentrated Mixtures

The yield of the coacervate complex was determined using a method described in previous studies [19]. The 0.2% (*w*/*v*) protein solution and four polysaccharide solutions were mixed at 1:1, 2:1, 4:1, and 10:1, respectively, adjusted to the maximum turbidity pH, and centrifuged at 10,000 rpm 4 °C for 10 min. The supernatant was discarded, and the precipitate at the bottom of the centrifuge tube was freeze-dried and weighed to calculate the yield. The yield of the coacervate complex was determined using the following equation:$$Yield=\frac{Quality\;of\;the\;complex\;coacervates\;on\;a\;dry\;basis}{Total\;mass\;of\;wall\;materials\;initially\;added}\times 100\%$$

### 2.7. Characterization of the Complex Coacervation Particles

FTIR experiments were performed using an FTIR spectrophotometer (Nicolet-460, Thermo Fisher Scientific, Waltham, MA, USA). The complexes obtained after freeze-drying were milled and pressed with potassium bromide and polymer in a ratio of 200:1. Transmittance values between 4000 cm^−1^ and 400 cm^−1^ were measured to determine the functional relationship between the different compounds.

XRD analyses of the samples were performed using a smart X-ray diffractometer (SmartLab 3KW, Rigaku Corporation, Tokyo, Japan), according to Dong et al. [20], with slight modifications. The prepared powder was flattened on a glass plate for X-ray diffraction analysis, and the scanning range was 2θ from 5o to 50°, with a step size of 5 °/min/scan rate.

Scanning electron microscopy (SEM) of the samples was conducted using a SU1510 SEM (Hitachi, Tokyo, Japan). The freeze-dried powder was adhered to a double-sided adhesive tape, coated with gold, and observed using a scanning electron microscope at a magnification of 500×.

An FEI Tecnai 12 (FEI Company, Hillsboro, NH, USA) was used for transmission electron microscopy (TEM). A drop of the complex coacervate solution was placed on a 400-mesh copper grid. The grids were then dried overnight at 40 °C, and the morphology of the samples was observed using an electron microscope with a single-tilt sample rack at an accelerating voltage of 100 kV.

### 2.8. Preparation of Pickering Emulsions Stabilized by Protein–Polysaccharide Complexes

SPI and the four polysaccharide suspensions were prepared by dispersing them in distilled water at 2% (*w*/*v*) and then hydrated for 24 h at 4 °C. The protein–polysaccharide complexes were mixed and adjusted to maximum turbidity. The protein–polysaccharide mixture was combined with soy bean oil at a ratio of 4:6 (*v/v*) (φ = 0.6). High-speed sheared on a homogenizer (FA25, FLUKO Company, Shanghai, China) at 8500 rpm for five min to obtain an emulsion. The apparent viscosities of the emulsions were measured using a rheometer (MCR102, Anton Paar, Graz, Steiermark, Austria). The mold plate was PP50, with a temperature of 25 °C, a gap of one mm, and a shear rate was 0.1–100 1/s. The linear viscoelastic region was measured in the 0.01–100% strain range. The obtained emulsions were subjected to an amplitude sweep test at 0.1–100 rad/s for 1 and 14 days to judge their stability.

### 2.9. Statistical Analysis

The data were presented as means ± standard deviation (*n* = 3) using SPSS 25.0 (SPSS Inc., Chicago, IL, USA). A one-way analysis of variance (ANOVA) with Duncan’s new multiple-range test was applied to analyze the significant differences between the means. Differences were deemed significant when (*p* < 0.05). The figures were designed using the Origin 2019 software (Microcal, MI, USA).

## 3. Results

### 3.1. Solubility of SPI as a Function of pH

The protein’s solubility is a significant characteristic that affects their emulsification, gel, and other tecno-functional properties during processing [1]. As shown in Figure 1, when the pH ranged from 2 to 11, the solubility of SPI exhibited a typical U-shaped curve, and the solubility was the lowest between pH 3 and 4. Yang et al. [5] reported that the isoelectric point of the obtained protein was four. The same protein molecules containing amino and hydroxyl groups were amphoteric molecules that existed as positive ions in the acidic medium and became negative ions beyond the isoelectric point. The protein exists in a zwitterionic state at its isoelectric point. The strong protein–protein connection leads to its aggregation and eventual precipitation, resulting in a minimum solubility near the isoelectric point. Olatunde et al. [1] stated that the solubility of proteins can be significantly affected by the amino acid structures, charges, and molecular weights. Above or below the isoelectric point value, owing to the existence of a charge on the polar group of the protein chain, the charges repel each other, the molecular dispersion is good, and the solubility gradually rises [21].

### 3.2. Turbidimetry of the Protein–Polysaccharide Complex Coacervate

Figure 2 shows the turbidity (absorbance at 600 nm) of SPI mixed with four polysaccharides at a total concentration of 0.2% (*w/w*) as a function of the pH at four different ratios. When the pH was gradually adjusted from 8 to 2, as reported in the literature [22], the change in the absorbance of the mixture from clarification to turbidity followed a two-step mechanism. First, a soluble complex formed at pH_c_, and the formation area of the soluble complex was visible with a decrease in the pH. The change in the solution from transparent to turbid and the rapid increase in turbidity at pHϕ1 resulted in the detection of the insoluble complex’s formation. Both macromolecules have a net negative charge at pH > pI, producing electrostatic repulsion between the chains in the protein–polysaccharide mixed system. Owing to the dilutive nature of the dispersion, biopolymers were considered to remain co-soluble rather than separate into two phases. During acidification, the charge on the protein’s surface began to combine with the protons when the protein was generally net positive, generating electrostatic attraction with the anionic polysaccharide chains. The formation of insoluble or soluble protein–protein and protein–polysaccharide complexes is usually caused by the electrostatic attraction between two biopolymers [23]. The solution became opaque and generated more insoluble electrostatic complexes and larger aggregates. When SPI was mixed with each of the four polysaccharides in different proportions, there was a double turbidity peak at 600 nm in the pH range of 8–2. Rapid acidification may cause the combination of the protonated positive amino groups (NH_3_^+^) of proteins with the negative carboxyl groups (–COOH) of polysaccharides, leading to a rapid decrease in the local pH and avoids the rearrangement or orientation of molecules [23]. The pH of the solution affects the surface group charges of the polymer, and the mixing mass ratio affects the charge balance of the biopolymer [24]. Therefore, these two parameters affect the strength of the protein–polysaccharide interaction and the complexity of the complex formation between them [22]. The change in pH_opt_ (the pH corresponding to the maximum turbidity) of the protein–polysaccharide polymer with different polymer blends and mass ratios is shown in Figure 2. When the SPI ratio increased, the pH_opt_ of the SPI–SA and SPI–CMC composite solutions increased. More SPI molecules were induced in the mixture to form electrically neutral complexes [22]. However, pH_opt_ did not change with the ratio in the SPI–GA mixture solutions. This phenomenon was also reported by Comunian et al. [8], who reported complex coacervation using gum Arabic and soluble pea proteins. The pH_opt_ values of SPI–CAR were steady at 4.5 at the ratios of 2:1 and 4:1 and increased to 5.0 at other ratios. In addition, the turbidity of the solution could be influenced by changing the size and quality of the aggregates; therefore, any change in the turbidity may be caused by the formation and decomposition of the protein–polysaccharide complexes [25].

### 3.3. ζ-Potential of SPI, Polysaccharides, and Their Complex Coacervates

Generally, the net surface charge can be determined using the ζ-potential. According to the DLVO electrostatic theory, the stability of the colloids is the balance between the van der Waals force attraction and the electric repulsion force [19]. The higher the absolute value of the ζ-potential, the stronger the electrostatic repulsion and the better the stability of the complex liquid. The moving particles’ dynamic electromotive forces were measured to understand better the electrostatic interactions between the four polysaccharides and SPI at different pH values (Figure 3).

As shown in Figure 3a–c, the ζ-potentials of SA, CAR, and CMC over the entire pH range were negative. An increase in the pH promoted the ionization of the carboxylic acid or sulfuric acid residues in the molecular structure of the polysaccharides, thus reducing the ζ-potential. The ζ-potential of the protein suspensions increases through the protonation of the carboxyl and amino groups during acid titration. The ζ-potential value of the protein suspensions increased through the protonation of the carboxyl and amino groups during acid titration. The ζ-potential of the sesame protein isolate solution changed from −19.1 to 19.9 as the pH was titrated from 7 to 2. The pH value at which the ζ-potential reached zero was considered the isoelectric point (PI), and the PI of SPI was approximately 4.6. However, because of the use of the lowest concentration of these proteins in the solution and the charge contribution of some impurities in the protein [25], the PI value of this protein, as determined by the potential value, was slightly higher than the previous absorbance value. The absolute value of the ζ-potential was affected by factors such as the polysaccharide type and concentration, and compared to the other polysaccharides, SA and CMC had higher negative charges. Notably, the ζ-potential of the mixture represented the net zeta value of the protein–polysaccharide complexes and non-interacting polymer monomers. With the decreasing pH, the ζ-potential of the complex increased. As the pH decreases, this is caused by electrostatic interactions between NH_3_^+^ in the protein polymer and the carboxyl groups on the polysaccharide.

With a smaller protein-to-polysaccharide ratio, the final ζ-potential of the complexes was lower, and the zero potential point moved to a lower pH value. When the charges of the two mixtures were almost neutralized, the protein–polysaccharide interaction was the strongest. When the charge density of the mix approaches zero, complexes form [26]. The turbidity of the solution is attributed to the appearance of scattering particles or droplets in the medium, which are related to the formation of aggregates [27]. Aggregation is mainly due to the lack of electrostatic repulsion between the biopolymers to form insoluble complex aggregates [19]. The more significant the proportion of protein, the more the protein molecules polymerize with polysaccharides in the biopolymer mixture to achieve the electrical neutrality of insoluble complexes. Therefore, negatively charged polysaccharide molecules interact with positively charged proteins to induce a charge reversal.

### 3.4. Coacervate Yield and Mean Particle Size at pH_opt_ of Complexes

Several investigations reported that the mixing ratio of the protein and polysaccharide is an important parameter affecting the complex aggregation because a suitable mixing ratio will lead to the whole interaction between the protein and polysaccharide molecules [28]. The turbidity of the system and coacervation yield are critical indicators for determining the optimal mixing ratio. The effects of the mixing ratio and polysaccharide on the coacervation yield and mean particle size of pH_opt_ are presented in Table 1. The coacervation yield of the SPI–GA was increased from 1:1 to 10:1. The higher yield indicated that SPI provided sufficient binding sites for the polysaccharides. More SPI caused self-aggregation, which increased the complex coacervation. A similar trend was observed in the mean particle size of the complexes at pH_opt_, which also increased with SPI content. When the protein content was relatively low, the main complex formed was soluble rather than insoluble. The low protein content reduces the electrostatic interactions with polysaccharides in complex mixtures [29]. The charge density, material utilization, and steric hindrance may affect the formation of complexes [19]. Plati et al. [21] also found similar results in a hemp protein isolate–gum Arabic complex. The highest yields of the SPI–SA, SPI–CMC, and SPI–CAR coagulants appeared at a ratio of 4:1, as shown in Table 1, which was also consistent with the turbidity shown in Figure 2. The low yield of the coagulants with a ratio of less than 4:1 might be due to the high polysaccharide content, which generates repulsive spatial forces, thus inhibiting complex coagulation and reducing turbidity. The maximum turbidity decreased when the mixing ratio exceeded 4:1, possibly because the increased SPI concentration led to self-interactions rather than interactions with polysaccharides.

### 3.5. XRD Analysis

Measuring the amorphous or crystalline state is significant for evaluating the physicochemical stability of the dried products [30]. The XRD analysis was used to examine the intermolecular and intramolecular interactions of the polymer networks. The results were used to determine the amorphous or crystalline properties of the proteins, polysaccharides, and other macromolecular biopolymers. The crystal structure is associated with the X-ray diffraction peak, whereas the amorphous form is associated with a broad peak. Figure 4 shows that pure SPI has two prominent diffraction peaks at 9.82° and 19.72°, which may be related to the existence of α-helices and β-folds in the polypeptide chain, respectively [31]. The pure GA diffraction pattern shows only a broad peak at 19.62°, indicating an electrostatic repulsion between the internal chains and an amorphous structure. CMC exhibited a sharp peak at 20° and was crystalline. The GA–SPI, CAR–SPI, CMC–SPI, and SA–SPI complexes all had a small and broad peak near 10° and a prominent diffraction peak near 20°, which indicated that the prepared polymers had a semi-crystalline structure. Compared to pure SPI and the four polysaccharides, the diffraction peak intensities of the polymers at 9° and 20° were significantly reduced, and the peak was broader.

The intensity of the X-ray diffraction peak reflects the grain size of the inner region of the crystal; the smaller the grain size, the lower the diffraction peak intensity [32]. In addition, the polymers prepared using SPI and the four polysaccharides exhibited two new sharp peaks at 32° and 45°. In addition, chia seed protein isolate–chia seed gum complex coacervates [33] indicated the existence of long-range crystallization peaks. The interaction between SPI and polysaccharides and the aggregation between polysaccharide chains promote intermolecular ordering [34]. This phenomenon also indicated that the protein and polysaccharide rearranged after the interaction, forming smaller amorphous condensates. Previous studies have shown that xanthan gum changes the crystalline properties of proteins by disintegrating the compact molecular arrangement of gelatin chains and interacting with gelatin [26,35]. The crystalline peaks of pectin disappeared when it was complexed with SPI, and the peak intensity of SPI was reduced by 20.1°. Hamedi et al. [31] found a lower diffraction peak intensity in the bovine serum albumin–cress seed gum complex coacervate than in the polymer alone. Eratte et al. [36] considered that amorphous structures were more easily dissolved and had stronger hygroscopicity.

Consequently, the core material can be released more easily from the capsules made from them. Owing to its crystalline nature, there were many sharp peaks in the XRD pattern of CAR, which was consistent with a previous study [37]. After the formation of the polymer with SPI, the characteristic CAR peaks almost disappeared. These results indicate that SPI molecules tightly adsorb the chain of CAR molecules, forming amorphous complexes between SPI and CAR through intermolecular interactions (electrostatic attraction and hydrogen bonding) [35], which is also consistent with the FTIR results.

### 3.6. FTIR Analysis

FTIR studies provide a better understanding of the mechanism of chemical interactions between biopolymers, resulting in the generation of new bonds and changes in the positions and intensities of characteristic peaks. Therefore, studying SPI’s conformational changes and the chemical bond stretching of SPI after forming complexes with SA, CAR, CMC, and SA is necessary. Figure 5 shows the FTIR spectra of SPI in a complex with four polysaccharides and their polymers. The bands at 2800–3000 cm^−1^ were related to the stretching mode of the C–H bond of methyl (–CH_3_), and the four polymers showed no notable differences compared to SPI. SPI exhibited characteristic protein absorption band peaks. Two peaks were found at 1658 and 1532 cm^−1^, related to the C=O stretching vibration of amide I, the C–N stretching vibration, and the N–H bending vibration of amide II, respectively [38]. The peak at 3100–3500 cm^−1^ is related to the –OH vibration, and the absorption band near 800–1200 cm^−1^ reflected the tensile vibration of C–C and C–O [9,39]. The complex condensations of SPI with the four polysaccharides were located at the 1646.64 and 1655.74 cm^−1^ peaks, relating to the non-esterified carboxyl group of the polysaccharide particles. The band at 1397 cm^−1^ represents the vibration of the C–N and N–H group planes of the amide III-bound amide or the vibration of the CH_2_ group of glycine. The peak at 1080 cm^−1^ was attributed to the C–O stretching vibration [26]. As shown in Figure 5d, the –OH stretching band (3387 cm^−1^) of GA and the –NH stretching band (3297 cm^−1^) of SPI shifted to 3301 cm^−1^ in the SPI–GA complex coacervates, indicating that a strong hydrogen bond was formed between the amide group of SPI and the hydroxyl group of GA during the aggregation process [30].

The peak at 1099 cm^−1^ moved to 1075 cm^−1^, and the intensity increased. The amide I peak changed from 1658 cm^−1^ to 1642 cm^−1^. This change may be due to the electrostatic interactions between the carboxyl group of GA and the amide group of SPI, combining the two vibrations in this region into a broader peak. This change may also be related to the transition of the protein from an α-helical structure to a more organized β-sheet and amorphous structure. In addition, the band intensities of amides I (1658 cm^−1^), II (1532 cm^−1^), and III (1237 cm^−1^) decreased and shifted slightly toward higher wavenumbers (amides I and III). This blue shift indicated that binding to the polysaccharide reduced the mobility of the amide [21]. This shift was caused by the electrostatic interaction between the C and N of the protein and polysaccharide ester group, which led to a change in the electron density. Therefore, the interaction between SPI and GA in the complex occurred mainly through the formation of hydrogen bonds and electrostatic interactions.

The peaks of CAR at 1637, 1247, and 1069 cm^−1^ correspond to the stretching vibration absorption signals of S=O, S–O, and C–O, respectively. In the SPI–CAR complex coacervates, the absorption peaks at 1247 cm^−1^ for CAR and 1312 cm^−1^ for SPI disappeared, whereas those at 1652, 1542, and 1069 cm^−1^ remained unchanged. In addition, a new peak was observed at 1724 cm^−1^.

It can be concluded that the interaction site lies between the N and H^+^ of NH_3_^+^ on SPI and the S–O^−^ of SO_3_^−^ on CAR [40]. Because the FTIR spectra of SPI and the four polysaccharide complexes differed for each biopolymer, it could be concluded that they were thermodynamically compatible and that intermolecular interactions occurred between the carboxyl groups on the polysaccharides and the functional groups of the SPI.

### 3.7. Rheological Property

The rheological properties affect the taste and stability of edible emulsions [41]. To compare the emulsifying characteristics of the SPI–SA, SPI–GA, SPI–CAR, and SPI–CMC complex coacervates, we measured the rheological characteristics of the emulsions at maximum turbidity pH, including the flow behavior and dynamic viscoelasticity of the emulsion. In oscillatory shear measurements, G′ is used for the storage modulus, G′ is used for the loss modulus, and tan δ (G″/G′) represents the loss coefficient, often used to characterize the viscoelastic characteristics of emulsions [11]. Figure 6a,b show the effects of the different shear rates and placement times on emulsion viscosity. It could be seen that the viscosity of each emulsion gradually decreased with the increase in the shear rate (0.1–100 s^−1^), which might be due to the fragmentation of oil droplets or shear decay of the complex structure [42], so all emulsions were a pseudoplastic fluid, and their apparent viscosity was reduced. When the shear rate was maintained within a specific range, the apparent viscosity of the emulsion samples varied with the different types of polysaccharides in the emulsion, and the emulsion samples with SPI–SA were the highest. After 14 days of storage, the apparent viscosity of the emulsion exhibited a similar trend, but the viscosities of all samples increased. This phenomenon may be due to the protein’s structural expansion and rearrangement at the oil–water interface after long-term storage caused by the dissipation flocculation of the excess complex or entanglement increase in the high-concentration complex [43]. These results indicate that the viscosity of the emulsion is not directly related to its stability during storage.

The oscillation test was performed in the linear viscoelastic region, as shown in Figure 6c,d. Within the measured frequency range, the G′ value was always higher than G″, indicating that these emulsions had high gel strength, and all emulsions expressed elastic characteristics. The emulsions prepared by adding SPI–SA, SPI–GA, and SPI–CMC had high energy storage moduli, enhanced with an increasing frequency.

The G′ and G″ of the SPI–CAR samples showed a weak frequency dependence, indicating that the applied deformation had no noticeable effect on the rheological response of the emulsion, which is consistent with the results reported by Jiang et al. [39]. Both moduli’s frequency dependence indicated that the mixture’s network primarily comprised noncovalent physical crosslinks [44].

The storage moduli (G′) of the SPI–CMC and SPI–SA emulsions at 0 and 14 days were much higher than the loss moduli (G′), resulting in the formation of smaller droplets with less aggregation and good stability, as shown in Figure 7, which may be related to the high spatial repulsion force between the droplets [43]. The droplets formed by the SPI–CAR emulsion were tiny and aggregated quickly. In contrast, the droplets formed by SPI–GA were large, easily aggregated during storage, and had poor stability, so their emulsion appeared to undergo phase separation (Figure 7).

### 3.8. Morphological Observation by TEM and SEM

To study the interaction between the sesame proteins and polysaccharides at the maximum turbidity pH, TEM was used to observe the microstructure of the complexes. SEM was used to analyze the surface structures of the freeze-dried complexes. Figure 8a,b showed that the TEM images of the SPI–GA and SPI–CAR polymers were spherically agglomerated. The TEM images of the SPI–CMC and SPI–SA polymers (Figure 8c,d) show dispersed spheres. The SEM images showed that freeze-dried SPI–GA and SPI–SA produced smooth and irregular sheets without cracks, whereas SPI–CAR and SPI–CMC showed irregular shapes, such as rods, sheets, and spheres. The weakly interconnected structures formed by the low-concentration solution collapsed due to the water’s evaporation during the lyze-drying. The microstructure of the protein–polysaccharide system is determined by the competition between the “phase separation” and “electrostatic interaction” between the proteins and polysaccharides [45].

The interaction between two large molecules with opposite charges creates a complex network of condensates that envelop the water molecules in the condensed phase, thus increasing the complexity of the structure.

## 4. Conclusions

The complex coacervation process depends highly on the pH and polysaccharide concentration. The optimum coacervation yield of the maximum turbidity of the SPI–GA mixture was at a ratio of 10:1 when the pH was 4.5. The SPI–GA mixture reached its maximum turbidity yield during coacervation at a ratio of 10:1, with a pH value of 4.5, representing the optimal conditions. Furthermore, as the biopolymer ratio decreased, the protein solubility curves shifted towards lower pH values, indicating a dependence on the pH. Concurrently, the FTIR and XRD analyses provided evidence of the formation between SPI and the four polysaccharides. During storage, the emulsion structures formed by SPI–CMC and SPI–SA demonstrated remarkable stability. In contrast, the emulsions of SPI–CAR and SPI–GA underwent breakage and fusion. The correlation between particle size and polymer shape was elucidated through particle size analysis, TEM, and SEM examinations. Electrostatic and hydrogen bond interactions were responsible for the binding of SPI with the four polysaccharides, with the polymer ratio and pH influencing the binding process. These analyses provided insights into the formation of a connected gel network structure by the complex. The current study’s findings enhanced our understanding of the interfacial interactions between SPI and mixtures of anionic polysaccharides. Ultimately, it is noteworthy that the SPI–CMC and SPI–SA complexes could be used as encapsulation agents for essential oils, probiotics, fat-soluble vitamins, and other active ingredients. Additionally, they could be applied to emulsion stabilizers, texture improvers, and fat substitutes.

## Figures and Tables

**Figure 1 foods-12-03696-f001:**
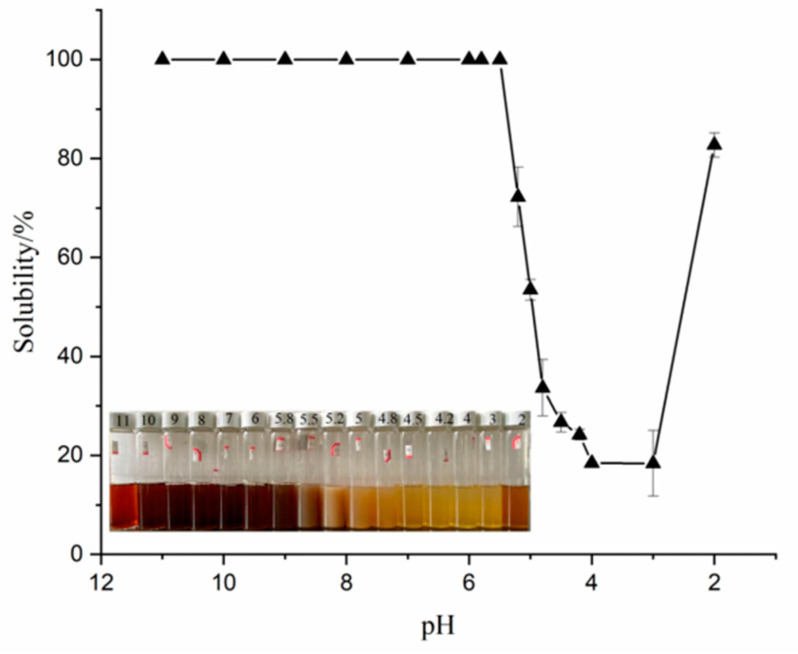
The solubility of SPI at different pH.

**Figure 2 foods-12-03696-f002:**
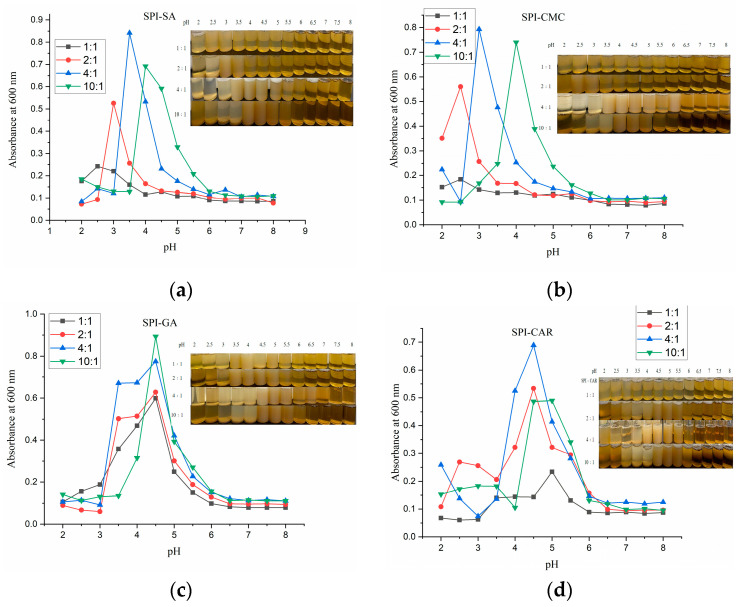
Turbidity of the SPI and four polysaccharide mixed solutions as a function of pH: (**a**) SPI–SA mixture solution; (**b**) SPI–CMC mixture solution; (**c**) SPI–GA mixture solution; (**d**) SPI–CAR mixture solution.

**Figure 3 foods-12-03696-f003:**
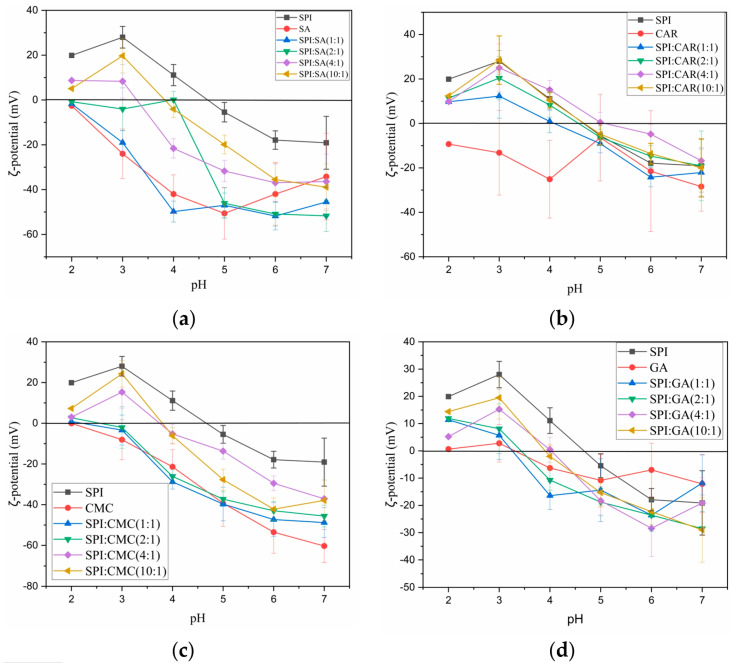
Mean ζ-potential (mV) of SPI and polysaccharides (SA, CAR, CMC and GA) and mixtures (SPI–SA (**a**), SPI–CAR (**b**), SPI–CMC (**c**) and SPI–GA (**d**)) at different ratios according to the pH.

**Figure 4 foods-12-03696-f004:**
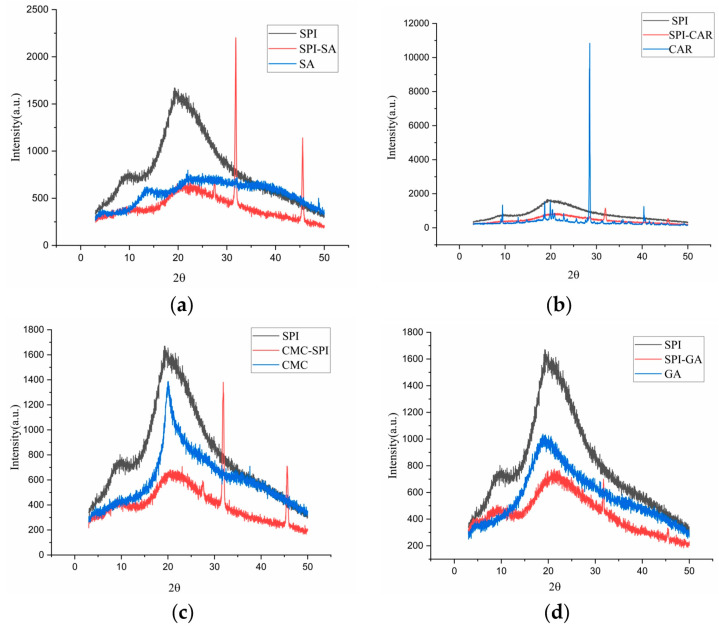
XRD of SPI, polysaccharides (SA, CAR, CMC and GA) and mixtures (SPI–SA (**a**), SPI–CAR (**b**), SPI–CMC (**c**) and SPI–GA (**d**)) at pH_opt_.

**Figure 5 foods-12-03696-f005:**
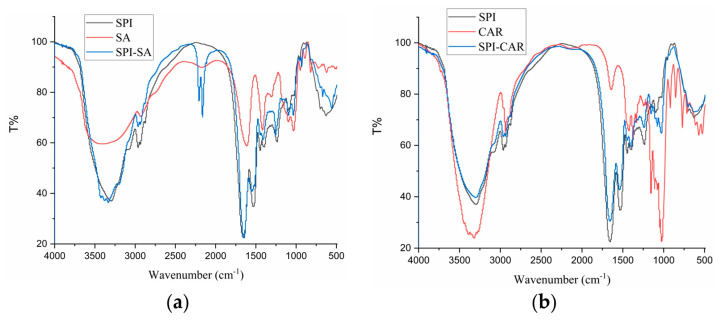
FTIR of SPI, polysaccharides (SA, CAR, CMC and GA) and complexes (SPI–SA (**a**), SPI–CAR (**b**), SPI–CMC (**c**) and SPI–GA (**d**)) at pH_opt_.

**Figure 6 foods-12-03696-f006:**
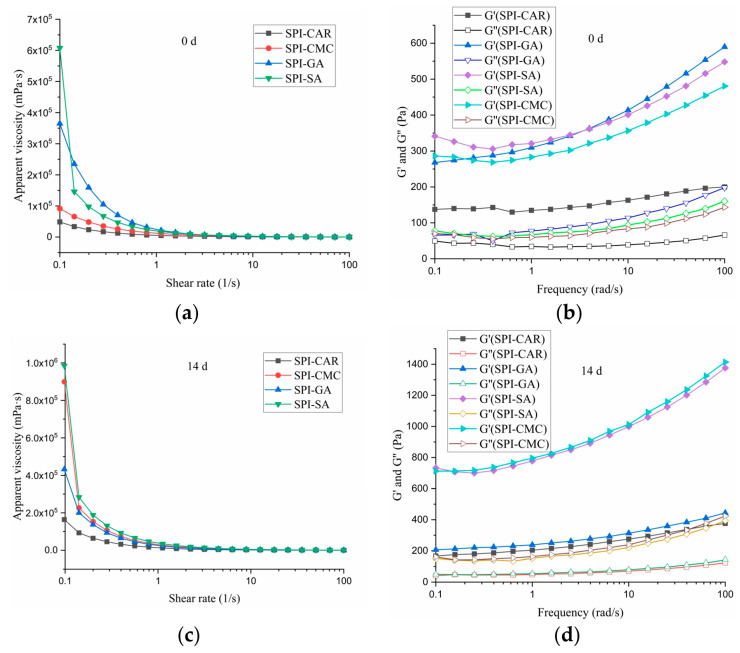
Rheological property of emulsion stabilized by protein–polysaccharide complex: appearent viscosity at 0 d (**a**) and 14 d (**c**), dynamic viscoelasticity of the emulsion at 0 d (**b**) and 14 d (**d**).

**Figure 7 foods-12-03696-f007:**
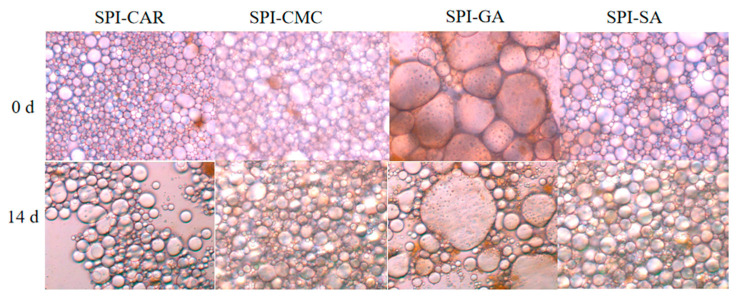
Optical microscopic images of emulsion stabilized by protein–polysaccharide complex at 0 d and 14 days.

**Figure 8 foods-12-03696-f008:**
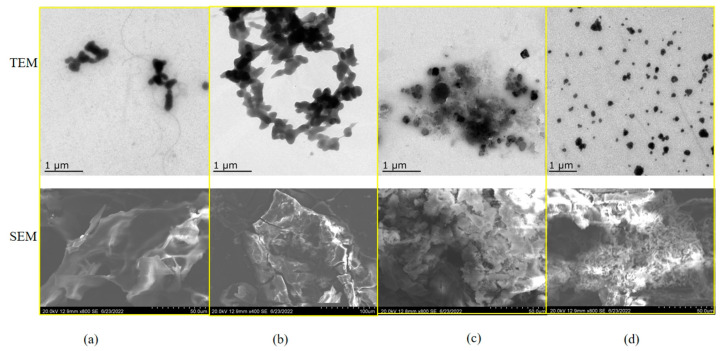
TEM image of the complex at pH_opt_ and SEM image of their lyophilization. (**a**) SPI–GA; (**b**) SPI–CAR; (**c**) SPI–CMC; (**d**) SPI–SA.

**Table 1 foods-12-03696-t001:** Coacervation yield and mean particle size at pH_opt_ of complexes.

	Coacervation Yield/%				Mean Particle Size/nm			
	1:1	2:1	4:1	10:1	1:1	2:1	4:1	10:1
SPI–GA	9.15 ± 0.78 ^D,b^	11.20 ± 0.14 ^C,c^	16.25 ± 0.64 ^B,d^	48.93 ± 2.12 ^A,a^	185.4 ^D,d^	196.8 ^C,c^	231.7 ^B,d^	265.6 ^A,d^
SPI–SA	10.70 ± 1.84 ^C,b^	22.75 ± 0.07 ^B,b^	51.64 ± 1.92 ^A,c^	10.25 ± 0.35 ^C,d^	297.0 ^C,b^	297.2 ^C,a^	397.6 ^B,a^	664.6 ^A,a^
SPI–CMC	7.85 ± 0.49 ^D,c^	58.90 ± 0.85 ^B,a^	69.75 ± 0.49 ^A,a^	36.57 ± 2.42 ^C,b^	311.0 ^C,a^	134.0 ^D,d^	365.6 ^B,c^	479.9 ^A,b^
SPI–CAR	17.30 ± 1.41 ^D,a^	57.05 ± 0.92 ^B,a^	62.93 ± 1.11 ^A,b^	30.40 ± 0.14 ^C,c^	253.2 ^C,c^	208.9 ^D,b^	376.0 ^B,b^	419.0 ^A,c^

The values were expressed as mean ± SD (*n* = 3). Uppercase superscripts indicate significant differences (*p* < 0.05) between treatments, while lowercase superscripts show differences within the same treatment.

## Data Availability

The data are contained within the article.

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
