# Peer review of "Fabrication and Characterization of Complex Coacervation: The Integration of Sesame Protein Isolate-Polysaccharides"

_foods, 2023, doi:10.3390/foods12193696_

Round 1

Reviewer 1 Report

In the presented article, the authors investigated the formation of insoluble complexes formed between sesame protein isolate and polysaccharide when mixed with a solution of biopolymers as a result of intermolecular interactions. Four different polysaccharides were used in the work namely arabic gum, carrageenan, sodium carboxymethyl cellulose, and sodium alginate. The influence of pH and the ratio of biopolymers on the properties of the resulting insoluble complexes was studied. The topic discussed in the article deserves attention, since polysaccharide-protein complexes formed as a result of non-covalent interactions have high potential for use as delivery systems for various ingredients in the food industry and in medicine. Despite the fact that quite a lot of work is currently devoted to this topic the authors managed to obtain new interesting data. The article was written logically, all conclusions are confirmed by the experimental data presented. While reading the manuscript, some minor comments and questions arose as follows.

1. The formula for determining the yield of the coacervate complex is not written (line 148)

2. What oil was used to prepare the emulsions?

Author Response

1. The formula for determining the yield of the coacervate complex is not written (line 148) Action: Done, See lines 149. 2. What oil was used to prepare the emulsions? Action: Done, we used soy bean oil to prepare the emulsions, I’ve added this in lines 174. Thanks for a lot for your time!

Reviewer 2 Report

In general, the manuscript is interesting and scientifically sound. Some aspects must be corrected:

Please adjust the sentence in line 78: “ The current (what?) will investigate….”

Carrageenan in line 80 is repeated twice.

If sesame is a protein-rich crop, how can we justify that the Kjeldahl method uses a conversion factor of 5.30? According to theory, a low conversion factor is used when proteins have low biological values, thus, we could have a rich-protein product but with low biological value.

Please, use a uniform manner to write “zeta potential”. Throughout the document are written “zeta potential” and “Zeta potential”

Figures a, b, c, and d, in Figure 6 are barely visible. It would be advisable to split this figure and create a new figure only including the images.

Please clarify the following sentence in lines 467-468: “Conversely, SPI-CAR and SPI-GA emulsions experienced”

In the introduction you mentioned the necessity of studying these phenomena to aid in the choice of polysaccharide stabilizers and the application of sesame protein in both the food and non-food sectors, however, in the conclusion, nothing is mentioned about applications in both sectors. Please include the utility of the results found in industrial applications.

Quality of English language should be performed in several paragraphs.

Author Response

1.Please adjust the sentence in line 78: “ The current (what?) will investigate….” Carrageenan in line 80 is repeated twice. Action: Done, See lines 79 and 80. 2. If sesame is a protein-rich crop, how can we justify that the Kjeldahl method uses a conversion factor of 5.30? According to theory, a low conversion factor is used when proteins have low biological values, thus, we could have a rich-protein product but with low biological value. Action: Thank you, sesame is a protein-rich crop, but it have a low biological value, we can see its biological value review in this paper https://doi.org/10.1155/2023/8577423, and the Kjeldahl ammonia determination method is not accurate and can only be roughly calculated, most edible beans and nuts choose 5.30 as a conversion factor. 3. Please, use a uniform manner to write “zeta potential”. Throughout the document are written “zeta potential” and “Zeta potential” Action: Done, Thank for your comment, all of “zeta potential” in this paper was change to “ζ-potential”. 4. Figures a, b, c, and d, in Figure 6 are barely visible. It would be advisable to split this figure and create a new figure only including the images. Action: Done, the modification was made based on the reviewer's comment. 5. Please clarify the following sentence in lines 467-468: “Conversely, SPI-CAR and SPI-GA emulsions experienced” Action: Done, See lines 470-472. 6.In the introduction you mentioned the necessity of studying these phenomena to aid in the choice of polysaccharide stabilizers and the application of sesame protein in both the food and non-food sectors, however, in the conclusion, nothing is mentioned about applications in both sectors. Please include the utility of the results found in industrial applications. Action: Done, Please see lines 479-482. Thanks for a lot for your time!

Reviewer 3 Report

The manuscript is well prepared, however needs to be improved before publication

Figure 1 the quality must be improved

Lines107-108 does the particle size measurement was observed?

The practical application of produced emulsions needs to be described

Author Response

1.Figure 1 the quality must be improved Action:Done 2. Lines107-108 does the particle size measurement was observed? Action: Done, please see the results in Table 1. 3. The practical application of produced emulsions needs to be described Action: Done. Please see lines 479-482 The practical application of the emulsions produced includes many tests. If these tests were included with this work, they would exceed the required limit of tables and figures and the number of words allowed according to the instructions of the Foods journal. These applications are now being prepared to be published in the Foods Journal later. Thanks for a lot for your time!